# Effect of Ionic Strength on Heat-Induced Gelation Behavior of Soy Protein Isolates with Ultrasound Treatment

**DOI:** 10.3390/molecules27238221

**Published:** 2022-11-25

**Authors:** Zhaojun Wang, Lin Zeng, Liwei Fu, Qiuming Chen, Zhiyong He, Maomao Zeng, Fang Qin, Jie Chen

**Affiliations:** 1State Key Laboratory of Food Science and Technology, Jiangnan University, Wuxi 214122, China; 2Analysis Centre, Jiangnan University, Wuxi 214122, China

**Keywords:** soy protein isolates, gel properties, NaCl concentration, ultrasound, rheological properties, protein interaction

## Abstract

This study investigated the effect of ultrasound on gel properties of soy protein isolates (SPIs) at different salt concentrations. The results showed that ultrasound could significantly improve the gel hardness and the water holding capacity (WHC) of the salt-containing gel (*p* < 0.05). The gel presents a uniform and compact three-dimensional network structure. The combination of 200 mM NaCl with 20 min of ultrasound could significantly increase the gel hardness (four times) and the WHC (*p* < 0.05) compared with the SPI gel without treatment. With the increase in NaCl concentration, the ζ potential and surface hydrophobicity increased, and the solubility decreased. Ultrasound could improve the protein solubility, compensate for the loss of solubility caused by the addition of NaCl, and further increase the surface hydrophobicity. Ultrasound combined with NaCl allowed proteins to form aggregates of different sizes. In addition, the combined treatment increased the hydrophobic interactions and disulfide bond interactions in the gel. Overall, ultrasound could improve the thermal gel properties of SPI gels with salt addition.

## 1. Introduction

Heat-induced gelation of plant proteins is essential for manufacturing plant-based meat and egg analogs [1]. Soy protein isolate (SPI) improves texture of gelation [2,3]; however, SPI has a much lower gel strength than myofibrillar protein or egg white protein [4]. Therefore, improving the heat-induced gelation of SPI has always been a topic of interest.

Several methods have been used to enhance the heat-induced gel properties of SPI, such as preheating, pH-shifting, transglutaminase cross-linking, and the addition of salt ions. Salt ions affect protein gelation, and the type and concentration of salt ions alter the final gel properties. Salt ions can change the charged state and the absolute net charge density of protein molecules and then modify the protein gelation process by affecting the aggregation kinetics and gel structure [5]. Salt ions alter the charge on the protein surface, the electrostatic repulsion, and the interactions of the protein with the solvent, thus altering the extension and aggregation of the peptide chain [6]. As the ionic strength increases, the absolute electrostatic charge density decreases, protein aggregation and gelation speed up, fractal aggregates in solution become more prominent and denser, and gel strength increases. Jiang et al. [7] observed that the addition of NaCl increased the turbidity of the soy protein solution more strongly at pH 7.0 during heating than in deionized water, indicating that aggregation is facilitated by shielding electrostatic repulsion. The increase in ionic strength accelerated protein aggregation. However, it has also been shown that the formation of large aggregates might increase the roughness of the gel structure and reduce the water-holding capacity (WHC) [8]. In addition, the high and low concentrations of salt ions would have different effects on the gel. Xu et al. [9] found that soy protein aggregation behavior strongly depends on the NaCl concentration. The size of protein aggregates in the solution increased and then decreased with the increase in ion concentration. Nakamura et al. [10] revealed that the protein forms a dense and ordered three-dimensional network structure at low ion concentrations, and a disordered and inhomogeneous gel structure at high ion concentrations. In addition, when salt ions were added to the protein solution, they would interact with the protein surface directly or indirectly, changing the stability and solubility of the protein, thereby affecting the gel properties [11,12].

However, excessive salt intake is not good for human health and increases the risk of high blood pressure and chronic diseases. Bohrer et al. [13] compared the sodium content in four meat analogue products ranging from 327.43 to 609.38 mg/100 g. Therefore, more attention should be paid to salt reduction in the manufacturing process of plant-based meat products.

Ultrasound treatment can be used to improve the gelation properties of plant proteins [14]. The cavitation effect generated by mechanical waves can cause structural changes, such as the reduction of protein aggregate size [15], the breaking of non-covalent bonds between proteins, and a change in the secondary, tertiary and quaternary structures of proteins [16]. These structural changes can improve the gelation of proteins. Hu et al. [17] found that ultrasound pretreatment resulted in a more dense and homogeneous acid-induced gel network, owing to smaller soy protein aggregates, higher surface hydrophobicity, and more soluble aggregates in solution. Tang et al. [18] used ultrasound technology to make insoluble aggregates in commercial soy protein-transformed soluble aggregates; the formation of soluble aggregates improved the thermally induced gelation ability of commercial soybean proteins. In addition, ultrasound combined heat treatment or glutamine aminotransferase catalytic treatment also enhanced gel properties [13,19]. Thus, the additional ultrasound treatment could be a potential method to reduce the salt added to a salty SPI gel.

This study investigated the effect of ultrasound treatment at different times on the thermal gelation of soy protein with varying NaCl concentrations. The relationship between protein surface charges, particle size, surface hydrophobicity, solubility, and heat-induced gel properties of soy protein was investigated. This study elucidates the effect of ultrasound on the changes in physicochemical properties of SPI at different salt concentrations and its effect on the performance of protein gels, which can provide a theoretical basis for the practical application of ultrasound in plant-based foods with different salt concentrations.

## 2. Results and Discussion

### 2.1. Particle Size Distribution

The particle size distribution can quickly reveal the degree of protein aggregation. As shown in Figure 1A, SPI without NaCl showed a bimodal distribution, ranging from 10 to 60 nm and from 60 to 400 nm. The particle size increased with NaCl concentration, indicating that salt ions caused SPI aggregation. This could be because salt ions decrease electrostatic repulsion and increase surface hydrophobicity [20]. At the same ion concentration, ultrasound treatment reduced the protein particle size, and increased ultrasound time further reduced the protein particle size. At 0–50 mM NaCl concentration, the protein particle size after ultrasound treatment was smaller than that of the control sample. At 100–600 mM NaCl concentration, the protein particle size after ultrasound treatment was larger than that of the control sample because the high ion concentration caused protein aggregation. The cavity and microfluidic beam effect generated during ultrasound treatment could change the protein interactions, such as enhancing the hydrophobic interactions between proteins, opening the tightly packed structure of protein aggregates, and reducing the protein particle size [21,22]. The combination of NaCl and ultrasound effectively modulated the size of protein aggregates, which was crucial to the properties of SPI gels.

### 2.2. ζ Potential

ζ potential is used to characterize the charged state of the protein surface. The surfaces of all proteins were negatively charged, and the ζ potential values were negative (Figure 2). As the salt concentration increased from 0 to 600 mM, the value of ζ potential increased from −42.9 mV to −22.8 mV, which was due to the electrostatic screening effect [20]. The ultrasound treatment showed little influence on the ζ potential value of SPI (*p* < 0.05), which was similar to faba bean protein reported by Alavi et al. [22]. An increase in the ζ potential value indicated a decrease in the amount of charge carried on the protein surface. In general, a low ζ potential value indicates electrostatic stability, and a high ζ potential values leads to protein aggregation [23]. Salt ions change the amount of charge on the surface of the protein, and the electrostatic repulsion force of the protein and the protein–solvent interaction force change with the number of charges, which, in turn, affects the gel process, including the extension and aggregation of peptide chains [6].

### 2.3. Surface Hydrophobicity

Surface hydrophobicity reflects the degree of protein development and exposure of surface hydrophobic groups, and these structural changes can alter the hydrophobic interactions between proteins, which, in turn, influence the formation of gel [24]. Figure 3 shows the effect of different treatments on the SPI surface hydrophobicity. The increase in NaCl concentration significantly increased the surface hydrophobicity of SPI (*p* < 0.05), and when the salt ion concentration was 200 mM, its surface hydrophobicity value reached 417, which was about twice that of the sample without salt. When the salt ion concentration increased further, the surface hydrophobicity value of the protein decreased. Combined with the particle size result (Figure 1), excess aggregates were formed in the solution, masking part of the hydrophobic groups on the protein surface and decreasing surface hydrophobicity. With the extension of ultrasound time, the hydrophobic groups on the surface of samples with different salt concentrations further increased. These results showed that more hydrophobic groups buried deep inside the protein were exposed when treated with ultrasound and salt ions, which increased the surface hydrophobicity of the SPI.

### 2.4. Protein Solubility

The solubility of proteins can reflect their structural change [25]. As shown in Figure 4, the solubility of SPI increased slightly from 87.89% to 89.20% in the presence of 50 mM NaCl and decreased with increasing NaCl concentration (100–600 mM). Ultrasound treatment significantly increased the solubility of soy protein (*p* < 0.05). These trends were consistent with Nazari et al. [26] and Hu et al. [27]. The smaller protein aggregates induced by the cavitation effect of ultrasound have higher protein–water interactions (Figure 1), leading to increased solubility [28]. Moreover, the strong physical force generated by the cavitation effect of ultrasound unfolded the protein structures. It exposed internal hydrophilic and polar groups to the protein surface, resulting in increased solubility of SPI [29].

### 2.5. Rheological Properties of SPI

The storage modulus (G′) and loss modulus (G″) as a function of temperature were used to characterize the rheological behavior of SPI during thermal treatment. As shown in Figure 5 and S1, the G′ values of SPI solutions decreased after increasing the temperature from 25 °C to 65 °C. When the temperature was above 65 °C, the G′ values rapidly increased with increasing temperature. These results suggested that the gel network structures began to form. In this stage, the hydrophobic groups embedded in the proteins were exposed and contributed to the formation of protein aggregates, which led to gel formation [30].

In the absence of NaCl, the ultrasound treatment had little influence on the G′ values of SPI gels (Figure 5A). The G′ values increased with the addition of NaCl up to 300 mM and decreased at 600 mM NaCl (Figure 5B–F). Adding NaCl at low concentrations could shield the surface charge of proteins and reduce intermolecular repulsion, facilitating gel formation. Adding more NaCl could inhibit the unfolding of the proteins and affect the gel network formation [31]. In the presence of NaCl, the G′ values increased with increasing ultrasound time, reaching a maximum when the sample was treated with 300 mM NaCl and 30 min ultrasound (Figure 5E). Ultrasound treatment increased the number of smaller protein particles (Figure 1), which could strongly aggregate with each other during the heating process, forming a more robust gel network [32]. Wang et al. [33] reported that the increase in surface hydrophobicity could also promote the enhancement of hydrophobic interaction between proteins so that protein gels form a denser gel network, which was conducive to the improvement of gel properties. In addition, good protein solubility provides better gel properties [2]. The increase in protein solubility facilitated the gel network formation and improved the protein gel properties. Therefore, the enhanced SPI gels treated with NaCl promoted protein aggregation by shielding the protein surface charge, while ultrasound reduced the aggregate size and increased the protein solubility and surface hydrophobicity.

### 2.6. Visual Appearance and Microstructure of Gels

Figure 6 shows the effect of NaCl and ultrasound treatment on the macrostructure and microstructure of SPI gels. Without NaCl, SPI formed a yellow opaque gel with little difference after ultrasound treatment (Figure 6A). The addition of NaCl increased the whiteness of the gels, which could be NaCl-facilitated SPI aggregation by shielding electrostatic repulsion, and was found to enhance the turbidity of the SPI solution.

The microstructure of the SPI gel without NaCl showed a uniform and smooth network structure with tiny pores (Figure 6B). With an increasing concentration of NaCl, the network structure of the gel became rough and uneven, and the pores became larger. After ultrasound treatment, a more uniform and compact three-dimensional network structure was presented. The protein gels with homogeneous and fine structures have high gel strength and WHC [34]. The changes in the microstructure of SPI gels were consistent with the trends in gel strength and WHC.

### 2.7. Texture Properties of Gels

Figure 7 shows the effect of the NaCl concentrations and ultrasound treatment on the gel properties of SPI. In the absence of NaCl, the hardness of SPI gels (~200 g) showed no significant difference with varying ultrasound treatment times (*p* > 0.05). The hardness of protein gels increased with increasing NaCl concentration up to 300 mM and decreased with more NaCl. The hardness of the protein gel had the highest value of 940 g at 300 mM with 30 min of ultrasound treatment, which was more than four times that of the untreated gel. This result indicated that adding NaCl in SPI with ultrasound treatment can enhance the gel network, in line with the rheological results (Figure 5). In addition, the sample with 50 mM NaCl and ultrasound treatment for 20 min (465 ± 9 g) had a similar gel hardness to the sample with 100 mM NaCl (523 ± 29 g). The gel hardness of the sample with 100 mM NaCl and ultrasound treatment for 30 min (735 ± 40 g) showed little difference to that of the sample with 200 mM NaCl (698 ± 66 g) (*p* > 0.05). Those results indicated that ultrasound was an effective salt reduction strategy for salty SPI gels.

### 2.8. Water-Holding Capacity of Gels

The WHC is closely related to the spatial structure of the gel. Figure 8 presents the WHC of the gels after ultrasound treatment at different times and NaCl concentrations. The WHC of the SPI gel was 100% when NaCl was not added; it decreased with the increase in NaCl concentration and dropped to a minimum of 95.36% (600 mM). The interactions between protein and water are an important factor affecting the WHC of the gel. The high concentrations of salt destroy the hydration layer on the protein surface and compete with the protein for water molecules, leading to a decrease in the WHC [35]. In addition, the salt ions caused SPI to form large aggregates, leading to larger pores in the gel network and a weaker ability to bind water. The WHC increased significantly with increasing ultrasound treatment time (*p* < 0.05). The ultrasound treatment reduced the particle size of SPI, allowing the gels to form a compact and homogeneous gel network. The compact and homogeneous microstructure could bind water in the protein matrix. Ultrasound treatment compensated for the loss of WHC caused by salt ions.

### 2.9. Chemical Interactions

To investigate the effect of NaCl addition and ultrasound treatment on the gel forces, we measured the solubility of soy protein gels in urea, β-ME, and SDS buffer. Both with and without the addition of 50 mM NaCl (Figure 9A,B), the gels had the highest solubility in urea. High concentrations of urea disrupt hydrogen bonding while weakening hydrophobic interactions. This indicates that hydrogen bonding and hydrophobic interactions were the main interacting force in the gels. The gel solubility in β-ME increased with ultrasound time (Figure 9A), suggesting that ultrasound promoted disulphide bond formation to form the gel network [36].

When 300 mM NaCl was added, the gel solubility in SDS was 10.19 mg/mL without ultrasound treatment and increased to 11.26 mg/mL with ultrasound treatment (Figure 9C). These results indicated that the salt ions converted the major interacting force from hydrogen to hydrophobic interactions, which contributed to the increase in the G’ value and gel hardness (Figure 4 and Figure 6). Li et al. [37] also found that adding salt ions promoted hydrophobic interactions in egg protein gels. The increased gel solubility in SDS indicated that ultrasound treatment benefits the hydrophobic interactions. This was consistent with the results of surface hydrophobicity (Figure 3). In addition, the gel solubility in β-ME was increased with the extension of ultrasound time; this increase implied an increase in disulphide bond interactions.

The gel solubility in all three solutions was significantly reduced at a NaCl concentration of 600 mM (*p* < 0.05) (Figure 9D). This is because the high concentration of salt ions promotes protein aggregation to produce insoluble aggregates while inhibiting the unfolding of protein structures. When compared to urea or β-ME, SDS still had the highest solubility. Thus, hydrophobic interactions remained the main force. Therefore, NaCl and ultrasound treatment improved the gel properties by changing the type and strength of interaction forces in the SPI gels and strengthening hydrophobic and disulphide bond interactions without reducing hydrogen interactions.

### 2.10. Pearson Correlation Analysis

NaCl addition and ultrasound treatment influenced the physicochemical and gel properties of SPI. The gel strength reached the highest value when the ζ potential was between −26 and −28, and ultrasound treatment further improved the gel properties. Therefore, when the ζ potential was −26 to −28, the Pearson correlation analysis was used to explore the relationship between the physicochemical and gel properties after ultrasound treatment. According to the particle size distribution, the SPI was divided into three parts: <106 nm (Size I), 106–1110 nm (Size II), and >1110 nm (Size III); the proportion of each part was calculated (Table A1). As shown in Table 1, the gel strength was significantly positively correlated with Size I, surface hydrophobicity, solubility, and WHC, and significantly negatively correlated with Size II and Size III (*p* < 0.05). There was a significant positive correlation between surface hydrophobicity, solubility, and WHC; all three were significantly positively correlated with Size I and significantly negatively correlated with Size II and III (*p* < 0.05). After ultrasound treatment, the surface hydrophobicity increased, the protein particle size was mainly concentrated in the Size I part, and the protein particles became small and uniform. Small protein aggregates enhanced protein–water interactions, resulting in increased solubility. In addition, the small protein particles facilitated the formation of a dense and uniform network structure, which locked in more water and increased the WHC of the gel. High solubility and a dense and uniform gel network contributed to gel strength. The Pearson correlation coefficient indicated that ζ potential, surface hydrophobicity, protein solubility, and particle size composition of SPI play essential roles in improving the gel properties.

### 2.11. Schematic Mechanism for Salt Ion and Ultrasound Treatment-Induced Changes of SPI Gels

The protein gels can be classified into two types according to the microstructure: opaque random aggregation type gel and transparent linear aggregation gel [36,37]. The gels in this study were random aggregation type based on the microstructure of SPI gels (Figure 6). A schematic diagram of the mechanism of improving the properties of SPI heat-induced gels by the combination of NaCl and ultrasound treatment is represented in Figure 10.

Native SPI aggregates have the smallest particle size and form a protein gel network with a small pore size and high water retention capacity, with hydrogen bonding as the main interacting force in the gel. After adding a certain amount of NaCl, the salt ions shield the electrostatic repulsion on the protein surface, and the charge on the surface is reduced, causing the SPI to aggregate and form large aggregates. The SPI forms an inhomogeneous gel structure during the gelation process, the network pore size becomes more prominent, and the main force of the gel changes from hydrogen bonding to hydrophobic interactions. Thus, the gel strength was improved, while the increased network pore size decreased the WHC. After ultrasound treatment, the size of SPI aggregates was reduced and was found to be more homogeneous. The ultrasound treatment increased the surface hydrophobicity and solubility, and enhanced the hydrophobic and disulphide bond interactions in the gel. At this time, the formed SPI gel structure was homogeneous, the network pore size was reduced, and the gel strength and water retention capacity were improved [38,39].

## 3. Materials and Methods

### 3.1. Materials

Soybean (Hong Feng No. 16) was purchased from Harbin Hong Yang Agricultural Development Co., Ltd. (Harbin, China). Sodium chloride (NaCl), sodium dodecyl sulfate (SDS), n-hexane, ethanol, and trichloroacetic acid (TCA) were purchased from Sinopharm Chemical Regent Co., Ltd. (Shanghai, China). Bovine serum albumin (BSA) and β-mercaptoethanol (β-ME) were purchased from Sigma-Aldrich Chemical Co. (St. Louis, MO, USA). All other chemicals were of analytical grade.

### 3.2. Preparation of Soy Protein Isolates

The SPI was prepared according to the procedure illustrated by Jiang [40]. The soybean was milled and sieved to obtain the soybean flour. To extract oil, the soybean flour was treated three times with n-hexane/ethanol (9:1, *v*/*v*). The defatted meal was dispersed in deionized water (1:10, *w*/*v*) and adjusted to pH 8.0 with 2 M NaOH. The dispersions were stirred at 25 °C for 2 h and centrifuged at 10,000× *g* at 4 °C for 20 min. The supernatants were adjusted to pH 4.5 with 2 M HCl, placed at room temperature for 30 min, and centrifuged at 3750× *g* for 10 min. The resulting precipitates were dissolved and adjusted to neutral pH of 7.0. The samples were freeze-dried and stored at −20 °C. The protein content of SPI was 93.53%, as determined by the Kjeldahl method (N × 6.25).

### 3.3. Ultrasound Treatment

The SPI dispersions (12%, *w*/*v*) were prepared with different NaCl solutions (0, 50, 100, 200, 300, and 600 mM) and stirred for 2 h at room temperature. An ultrasound processor (JY92-IIN, Scientz Biotechnology Co. Ltd., Ningbo, China) with a 6 mm diameter amplitude transformer was used to treat 40 mL of SPI solution. The amplitude transformer was placed 1 cm below the solution during the treatment process. The protein solution was placed in ice water to avoid overheating. Ultrasound treatment was performed in the pulsed mode (on-time 5 s; off-time 5 s) at a frequency of 20 kHz and power of 300 W for 0, 5, 10, 20, and 30 min. The ultrasound intensity was 25 W/cm^−2^, based on the Zhong and Xiong method [41].

### 3.4. Particle size and ζ Potential Analysis

The particle size distribution and ζ potential of the protein samples were measured using a Ζsizer Nano ZS instrument (Malvern Instruments, London, UK). All samples were diluted with deionized water to 1 mg/mL.

### 3.5. Surface Hydrophobicity

The surface hydrophobicity was measured according to the method demonstrated by Kato and Nakai [42] with some modifications. The protein samples were diluted with phosphate buffer (10 mM, pH 7.0) to prepare protein solutions with the following concentrations: 0.02, 0.04, 0.06, 0.08, and 0.1 mg/mL; 2 mL of the protein samples and 20 μL of ANS (8.0 mM in 10 mM phosphate buffer, pH 7.0) were mixed and kept for 3 min in the dark at room temperature. The fluorescence intensity (FI) was measured using a spectrofluorometer (F-2700, Hitachi Co., Japan) with an excitation wavelength of 390 nm and an emission wavelength of 470 nm. The slit width was 5 nm. The initial slope of the plot of FI versus the protein concentration was calculated by linear regression analysis and used as an index of protein hydrophobicity (H0).

### 3.6. Protein Solubility

The protein solubility was measured according to the method demonstrated by Liang et al. [43] with some modifications. The protein solution (1%, *w*/*v*) was dispersed in deionized water. The dispersions were mixed using a magnetic stirrer at ambient temperature for 1 h and centrifuged at 12,000× *g* for 15 min. The Biuret method used BSA as a standard to determine the soluble protein content. Protein solubility was calculated as the percentage of the protein in the supernatant over the total protein in the aqueous protein solution.

### 3.7. Low-Amplitude Oscillatory Measurements

The low-amplitude oscillatory measurements were carried out using a Haake Mars Ⅲ rheometer (Thermo Electron GmbH, Dieselstr, Karlsruhe, Germany), referring to the method described by Schmidt et al. [44] with some modifications. A circular parallel-plate geometry (35 mm diameter) with a gap of 1 mm was used between the plates. Strain (0.01–10%) and frequency sweep (0.01–10 Hz) tests were performed to determine the linear viscoelastic region at 25 °C. A temperature sweep test was performed with 1 Hz and 1% strain frequency. The SPI solutions were heated from 25 °C to 95 °C at a rate of 2 °C/min, held at 95 °C for 10 min, and cooled from 95 °C to 25 °C at the same rate. Silicon oil was added to the edge of the sample to prevent moisture evaporation during the measurements.

### 3.8. Heat-Induced Gel Preparation

The SPI solution prepared in 3.3 was transferred into a 10 mL beaker (25 × 39 mm, diameter × length) and sealed with plastic wrap. All beakers were immersed in a water bath at 95 °C, maintained for 35 min, and immediately cooled in ice water. All gel samples were stored at 4 °C overnight.

### 3.9. Texture Analysis of Gels

The gel texture was determined using a TA-XT2 texture analyzer (Stable Micro Systems, Ltd., Godalming, UK), referring to the method described by Lv et al. [45] with some modifications. After 1 h of equilibration at 25 °C, the protein gels were carefully removed from the beaker and cut into cylinders of 25 × 20 mm (diameter × length). The P/36R probe was selected to measure with the following test parameters: pre-test speed of 1 mm/s, test speed of 5 mm/s, post-test speed of 5 mm/s, 5.0 g trigger force, 50% compression deformation, a two-cycle compression test, and a 5 s time interval between two cycles.

### 3.10. Water-Holding Capacity of Gels

The WHC was tested by following the procedures demonstrated by Min et al. [46] with slight modifications. Briefly, SPI gels were weighted as W_1_ and centrifuged at 10,000× *g* for 15 min at 4 °C. The samples after centrifugation were weighted as W_2_. The WHC was calculated as follows:$$\mathrm{WHC}\;\left(\%\right)=\frac{W_{1}-W_{2}}{W_{1}}\times 100\%$$

### 3.11. Microstructure of Gels

The microstructure of the gels was observed using scanning electron microscopy. The prepared protein gels were frozen in sufficient liquid nitrogen (−196 °C) and lyophilized. The dry gold-sputtered samples were observed under a SU8100 electron microscope (Hitachi, Tokyo, Japan) in high vacuum mode with an acceleration voltage of 3 kV.

### 3.12. Chemical Interactions of Gels

The chemical interactions were measured according to Jiang and Xiong [7]. Two grams of SPI gel samples were blended with 18 mL of three different buffers and homogenized with a Model Ultra-Turrax18 homogenizer (IKA Works GmbH & Co., Staufen, Germany) at 13,500 rpm for 10 s. The homogenized solution was held in a water bath at 80 °C for 1 h and centrifuged at 10,000× *g* for 15 min. The supernatant was used to determine the protein content by the Biuret method using BSA as a standard. The three different buffers used were: 50 mM phosphate with 8 M urea (pH 7.0), 50 mM phosphate with 0.5% SDS (pH 7.0), and 50 mM phosphate with 0.25% β-ME (pH 7.0). All solutions containing β-ME were used to precipitate proteins with 12% TCA to avoid β-ME interfering with the absorbance. Before the Biuret method determined the protein content, the precipitated proteins were re-solubilized with 0.5 M NaOH.

### 3.13. Statistical Analysis

All the experiments were performed in triplicate. The data were expressed as the mean ± standard deviation (SD). SPSS version 26.0 (IBM, Armonk, NY, USA) was used for the analysis. A *p*-value of <0.05 indicated a significant difference. The Pearson correlation analysis was carried out, and the statistical significance was defined at *p* < 0.05 and *p* < 0.01 by a two-tailed test.

## 4. Conclusions

The effect of ultrasound treatment and NaCl addition on the gel properties of SPI gels was investigated, and the mechanism to enhance the gel hardness was explored. At 200 mM NaCl with 20 min of ultrasound treatment, the hardness of the protein gel was four times that of the untreated gel. Without NaCl addition, the ultrasound treatment improved surface hydrophobicity and solubility and reduced particle size but did not enhance gel hardness. In the presence of NaCl (50–600 mM), ultrasound treatment changed the aggregate particle size. It significantly increased the ζ potential, surface hydrophobicity, and solubility, improving the gel network structure and enhancing gel hardness and WHC. NaCl addition and ultrasound treatment changed the main interacting gel forces from hydrogen bonding to hydrophobic interactions and strengthened the hydrophobic and disulphide bond interactions without reducing the hydrogen bonding. Therefore, combining ultrasound treatment and NaCl addition is an effective method to enhance the properties of SPI heat-induced gels.

## Figures and Tables

**Figure 1 molecules-27-08221-f001:**
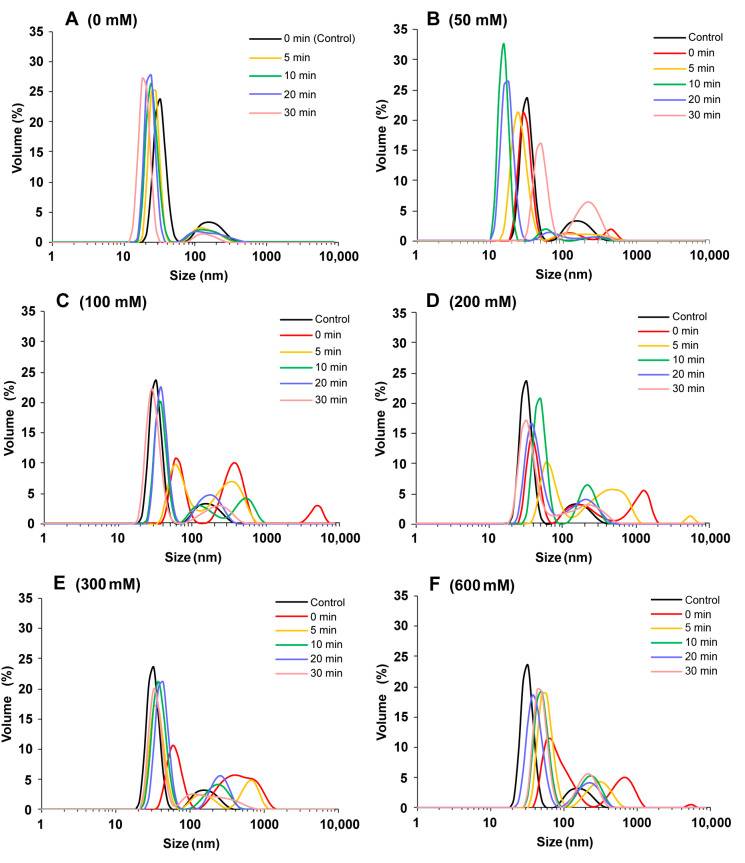
Particle size distribution of soy protein isolate treated with varying ultrasound processing time (0, 5, 10, 20, and 30 min) at different NaCl concentrations at 0 mM (**A**), 50 mM (**B**), 100 mM (**C**), 200 mM (**D**), 300 mM (**E**), and 600 mM (**F**).

**Figure 2 molecules-27-08221-f002:**
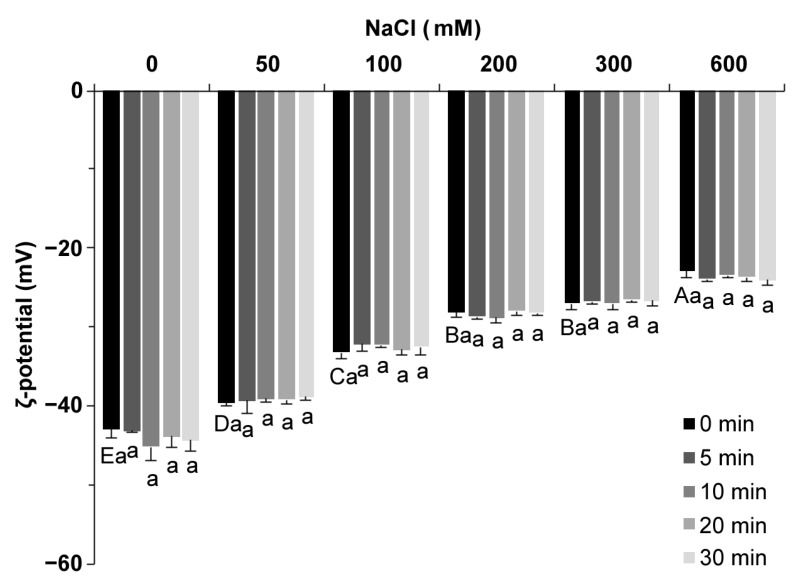
ζ potential of soy protein isolate treated with varying ultrasound processing times (0, 5, 10, 20, and 30 min) at different NaCl concentrations (0, 50, 100, 200, 300, and 600 mM). The upper letters indicate significant differences between the samples without ultrasound treatment in different salt concentrations. The lower letters indicate significant differences between the samples with different ultrasound times in the same salt concentration (*p* < 0.05). Different letters indicate significant differences between groups (*p* < 0.05).

**Figure 3 molecules-27-08221-f003:**
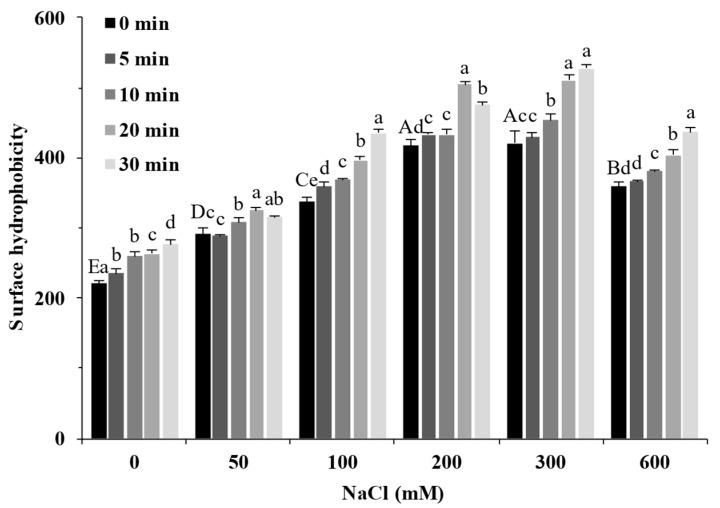
Surface hydrophobicity of soy protein isolate treated by ultrasound at different NaCl concentrations for different processing times. The upper letters indicate significant differences between the samples without ultrasound treatment in different salt concentrations. The lower letters indicate significant differences between the samples with different ultrasound times in the same salt concentration (*p* < 0.05).

**Figure 4 molecules-27-08221-f004:**
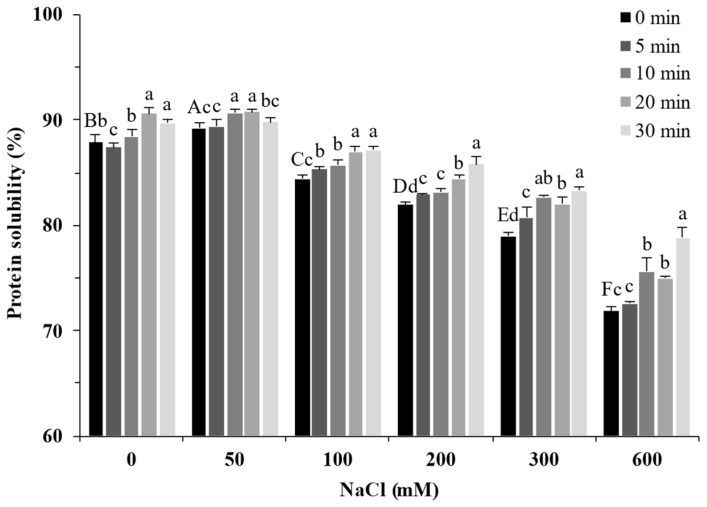
Solubility of soy protein isolate treated by ultrasound at different NaCl concentrations for different processing times. The upper letters indicate significant differences between the samples without ultrasound treatment in different salt concentrations. The lower letters indicate significant differences between the samples with different ultrasound times in the same salt concentration (*p* < 0.05).

**Figure 5 molecules-27-08221-f005:**
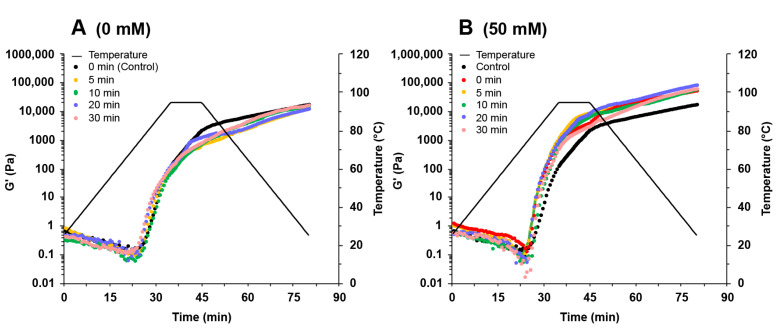
Change in storage modules (G′) during the heating–cooling cycle of soy protein isolate treated by varying ultrasound processing times (0, 5, 10, 20, and 30 min) at different NaCl concentrations at 0 mM (**A**), 50 mM (**B**), 100 mM (**C**), 200 mM (**D**), 300 mM (**E**), and 600 mM (**F**).

**Figure 6 molecules-27-08221-f006:**
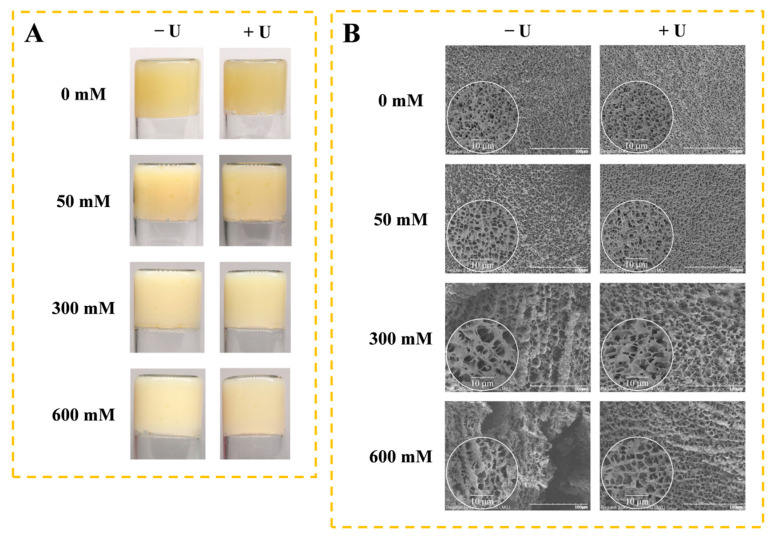
Macrostructure (**A**) and microstructure (**B**) of soy protein isolate gels as influenced by different concentrations of NaCl with ultrasound treatment. − U: SPI without ultrasound treatment; + U: SPI with 30 min of ultrasound treatment.

**Figure 7 molecules-27-08221-f007:**
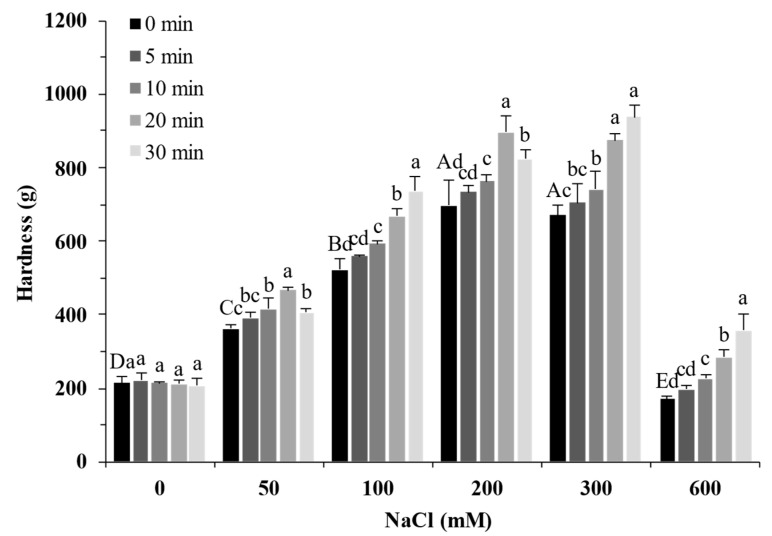
Gel hardness of soy protein isolate treated by varying ultrasound time at different NaCl concentrations. The upper letters indicate significant differences between the samples without ultrasound treatment in different salt concentrations. The lower letters indicate significant differences between the samples with different ultrasound times in the same salt concentration (*p* < 0.05).

**Figure 8 molecules-27-08221-f008:**
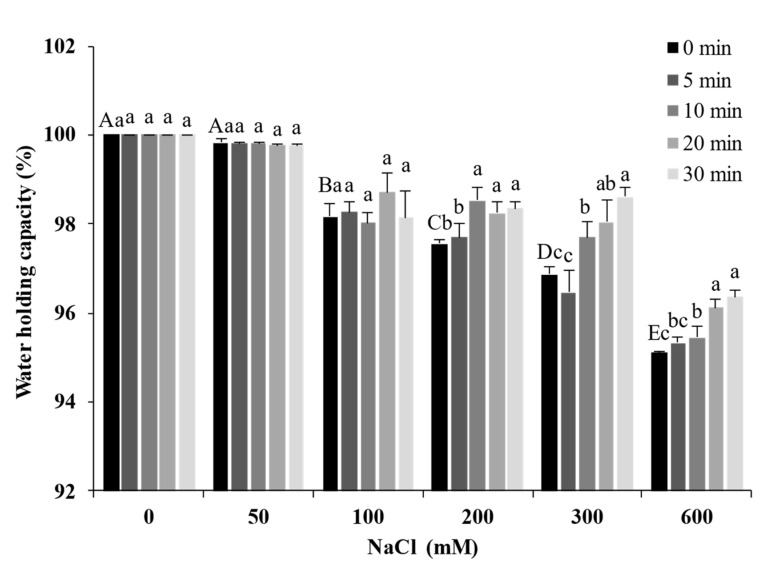
Water-holding capacity of soy protein isolate treated with ultrasound at different NaCl concentrations for different processing times. The upper letters indicate significant differences between the samples without ultrasound treatment in different salt concentrations. The lower letters indicate significant differences between the samples with different ultrasound times in the same salt concentration (*p* < 0.05).

**Figure 9 molecules-27-08221-f009:**
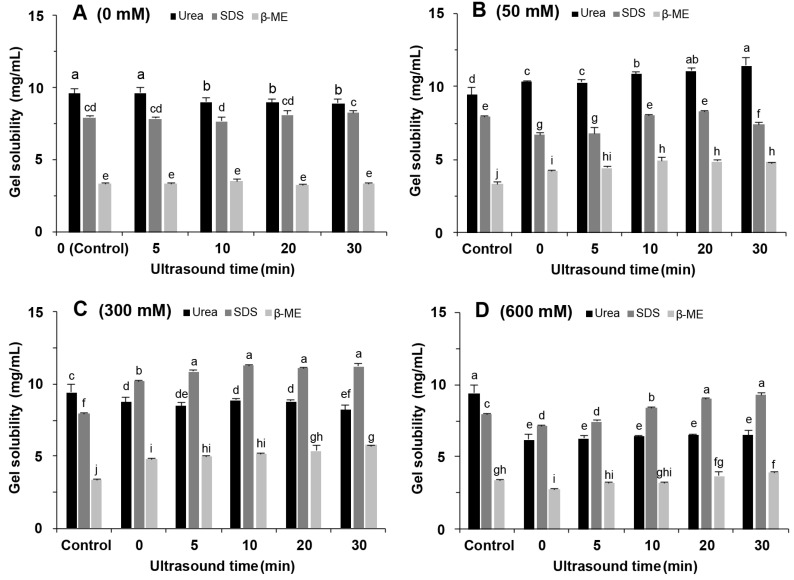
Solubility of soy protein isolate gel in different solvents. (**A**) 0 mM NaCl; (**B**) 50 mM NaCl; (**C**) 300 mM NaCl; (**D**) 600 mM NaCl. Different letters indicate significant differences between groups (*p* < 0.05).

**Figure 10 molecules-27-08221-f010:**
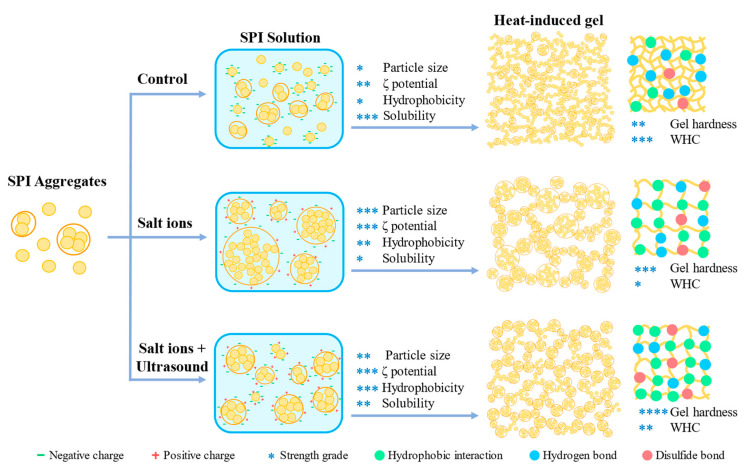
Schematic diagram of the mechanism of improving soy protein isolate heat-induced gel properties by combining NaCl addition and ultrasound treatment. *, **, *** and **** represent the values of the test index (particle size, ζ potential, hydrophobicity, solubility, gel hardness and WHC).

**Table 1 molecules-27-08221-t001:** Correlation analysis. * and ** indicate significant levels at *p* < 0.05 and *p* < 0.01, respectively. Size I represents a protein with a volume fraction less than 106 nm, Size II represents a protein with a volume fraction of 106 to 1110 nm, and Size III represents a protein with a volume fraction greater than 1110 nm (the specific data is given in Table A1).

	Size I	Size II	Size III	ζ Potential	Surface Hydrophobicity	Protein Solubility	Gel Strength	WHC
Size I	1							
Size II	−0.998 **	1						
Size III	−0.832 **	0.794 **	1					
ζ potential	−0.047 **	0.04	0.096	1				
Surface hydrophobicity	0.704 **	−0.719 **	−0.452 *	0.323	1			
Protein solubility	0.716 **	−0.723 **	−0.523 **	0.239	0.502 **	1		
Gel strength	0.748 **	−0.738 **	−0.692 **	−0.387 *	0.908 **	0.592 **	1	
WHC	0.687 **	−0.697 **	−0.471 **	−0.256	0.581 **	0.704 **	0.679 **	1

## Data Availability

The data presented in this study are available on request from the corresponding author.

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
