# Peer review of "Effect of Ionic Strength on Heat-Induced Gelation Behavior of Soy Protein Isolates with Ultrasound Treatment"

_molecules, 2022, doi:10.3390/molecules27238221_

Round 1
Reviewer 1 Report
The manuscript described the different NaCl concentrations and ultrasound treatment on the heat-induced gel properties of soy protein isolates. My specific comments are shown as follows:
Abstract: it should also show the main finding from the research results, especially the numerical data. In Line 9, the reader may hard to understand the ζ potential.
In the introduction, the novelty of your current study (compared to other relevant studies) should be clarified
Line 91: The discussion of ζ. potential is rather weak, please discuss it deeply.
Figures 2, 3, 6, and 7: it is hard to understand the meaning of the capital in the figures. Please state clearly what the upper letters and lower letters mean.
Line 171: "P>0.05 should be added after the sentence if there is no significant" Please check throughout the manuscript.
Conclusion: so what the best NaCl addition and ultrasound treatment time for obtaining good gel properties should be concluded here.
Reviewer 2 Report
Comments and Suggestions for Authors
General comments
This manuscript proposes the effect of ultrasound treatment and NaCl addition on the heat-induced gel properties of soy protein isolates. In order to improve the SPI gel hardness and WHC, NaCl addition and ultrasound treatment were used in this research the novelty of the work. In addition, this manuscript described that SPI gel hardness could be improved by changing the aggregate size, protein solubility, and the gel’s type and strength of interacting forces.In general, this is a high-quality article with complete data and a very logical discussion of the data.
However, before publication, some aspects need to be clarified and the manuscript improved. Also, a careful revision of the text is required.
Abstract
1. The influence of the best conditions (ultrasound treatment and NaCl addition) on the gel properties and water holding capacity of protein gel were not described in the abstract, and adding specific values is more conducive to reading.
Introduction
2. Line 70: Ultrasonic technology is a common method to improve gel. Is there any effect of ultrasound combined with other technologies (like heating treatment) on gel.
Results and discussion
3. The particle size of protein in Fig. 1A-B decreased with ultrasonic treatment time, while the particle size in Fig. 1C-F increased. The statement in Section 2.1 was inaccurate.
4. Section 2.2: Please describe more articles published about the relationship between ζ potential and gel to understand this part
5. The temperature ordinate of Fig. 4D was different from that of other figures in Fig.
6. Section 2.5: It is more scientific to use color difference value (whiteness) to explain.
7. Section 2.8: “Without or with 50 mM of NaCl addition (Fig. 8A, B), the gels had the highest solubility in urea”. Urea can not only break the hydrogen bond but also break the hydrophobic interaction. Is it stated that the hydrogen bond and hydrophobic interaction are the main interaction forces under the condition of low NaCl concentration?
8. Section 2.9: The author divided the particle size into three parts: <106 nm (Size I), 106-1110 nm (Size II), and >1110 nm (Size III). What is the basis?
9. The specific particle size of Size I, Size II, Size III should be indicated in Table 1 to help better understand
Materials and methods
10. Section 3.3: Is the setting range of salt ion concentration too large? Because so much salt is usually not added in the process of food processing
11. Section 3.3: Why did the author choose the ultrasonic treatment condition at 300W, 20Hz. The author chose to treat the samples with different ultrasonic time under the same ultrasonic power, and whether to consider using different ultrasonic power to research.
12. Section 3.6: Is silicone oil used in the strain and frequency sweep? It is reasonable that silicone oil was not to use at low temperature. How much does the use of silicone oil affect the rheology of protein at low temperature
13. Section 3.7: Please describe the size of the beaker
14. Section 3.8: Mark references
15. Section 3.8: How to cut a cylinder (25×20 mm)?
16. Section 3.11: “The homogenized solution was held in a water bath at 80°C for 1 h”, whether this operation would cause interaction between proteins.
17. Line 336: The unit of urea concentration is wrong, it should be 8mol/L or M.
Conclusions
18. Line 352: “It significantly increased ζ potential and solubility, improving the gel network structure and enhancing gel hardness, springiness, and WHC”. There is no discussion about springiness in the article.

Reviewer 3 Report
This work focus on the effects of ion strength and ultrasound on the gelation behavior of soy protein isolate. There is several similar work on pea protein, beta-lactoglobulin, and fibrial protein. I have not find out the novelty of this work.
Major:
1. line 287: what does it mean by neutral pH? the formation of NaCl during pH adjustment has been ignored in this work.
2. Line 110: this paper stated that the ultrasound treatment lead to exposion of internal hydrophilic and polar groups to the protein surface. There is no data to support it. Actually, hydrophobic area is more easier to expose out side durting heat or pH treatment.
3. Line 85: the combination of NaCl and ultrasound effectively modulated the size of protein aggregates. This conclusion was drawn based on which data. I can not get this conclusion from Figure 1.
Based on these, I think the authors have not elucidate the results and conclusions based on their data.
Round 2
Reviewer 1 Report
The authors have addressed my comments. Next time, please answer the reviewers' comments with the Line number in the revised manuscript
Reviewer 3 Report
it can be accepted.